# OpenReview forum: "Boosting Reinforcement Learning with Extremum Experiences"
_ICLR.cc/2024/Conference — Submitted to ICLR 2024_

### Official Review · Reviewer_uoX8 · 2023-10-26

**Soundness:** 2 fair
**Presentation:** 3 good
**Contribution:** 2 fair
**Rating:** 5
**Confidence:** 4

**Summary:**

This work proposes a new algorithm, MaxMin TD Learning, by modifying $\epsilon$-greedy exploration in DQN. Specifically, with probability $\epsilon$, the argmin action is selected given the state-action values. Theoretically, this leads to higher temporal difference error under certain assumptions. In practice, the proposed algorithm is shown to achieve higher sample efficiency than DQN with $\epsilon$-greedy exploration.

**Strengths:**

As far as I know, the presented idea is novel and easy to implement. Generally, the paper is easy to follow. The advantage of the proposed algorithm is supported by both theories and experiments. All algorithms are tested in 100K Atari games.

**Weaknesses:**

The major weaknesses are insufficient experiments, a gap between theory and experiments, and a lack of explanation.

- How large is $\mathcal{D}(s)$, $\delta$, and $\eta$ in practice? Is $\mathcal{D}(s) − 2\delta − \eta$ positive or negative in practice?
- In Section 4, a fixed step size is used. How does the performance of the algorithms vary with different step sizes?
- In Section 4, $\epsilon$ is chosen from $[0.15, 0.25]$. In practice, a smaller $\epsilon$ is usually used. How is the performance of the algorithms with smaller $\epsilon$, such as $\epsilon \in [0.01, 0.05]$? How sensitive is MaxMin TD learning to $\epsilon$ compared to DQN?
- In Figure 4, not all tasks (e.g. Amidar, Bowling, BankHeist, and StarGunner) are trained with 200M frames although it is claimed so.
- In Section 3, it is claimed that, in the early phase of the training, in expectation over the random initialization $\theta \sim \Theta$, the TD error is higher when taking the minimum value action than that of a random action. However, this contradicts the experimental results shown in Figure 3, especially Figure 3(a).
- Lack of explanation: Why would a higher TD error help exploration and speed up training in general? I don't see a clear connection between them. I believe that it is very important to explain the logic behind.

**Questions:**

- In Definition 3.2 & 3.5: What is $\hat{a}(s,\theta)$?
- In Proposition 3.4, $a_t \sim \mathcal{U},(\mathcal{A})$: typo.
- In Section 4, it is mentioned that the maximum achievable reward in 100 steps is 10. However, the learning curve in Figure 1 (b) is above 10 in the end. How do you get the data used in Figure 1?

---

> ### Author Response · Authors · 2023-11-21
> **Author Response**
>
> Thank you for allocating your time to prepare a response for our paper.
>
> **1.** *”Lack of explanation: Why would a higher TD error help exploration and speed up training in general? I don't see a clear connection between them. I believe that it is very important to explain the logic behind.”*
>
> The TD error is the loss function used in Q-learning i.e. the goal is to minimize the TD error over all state transitions. By focusing on state transitions where the TD error is higher, larger updates can be made to the Q-function, resulting in faster learning. To give an extreme example, if the TD error for a transition is zero, then the gradient of the loss is zero, and using that transition to make an update results in zero change to the Q-function (i.e. zero learning).
>
> **2.** *”In Section 4, $\epsilon$ is chosen from [0.15,0.25]. In practice, a smaller $\epsilon$ is usually used. How is the performance of the algorithms with smaller $\epsilon$, such as [0.01,0.05]? How sensitive is MaxMin TD learning to compared to DQN?“*
>
> We now added a new section in the supplementary material regarding your question. The results demonstrate that MaxMin TD learning also performs substantially better with $\epsilon= [0.01,0.05]$ as you suggested. Please see supplementary material.
>
> **3.** *“How large is $\mathcal{D}(s)$, $\delta$ and$\eta$ in practice? Is $\mathcal{D}(s)-2 \delta -\eta positive or negative in practice?”*
>
> In practice $\mathcal{D}(s) - 2\delta -\eta$ is positive as demonstrated by our empirical results on the temporal difference error of MaxMin TD learning versus the $\epsilon$-greedy in Figure 3 and Figure 6. In particular, the higher TD values for MaxMin TD learning demonstrate that this quantity is indeed positive in practice.
>
>
> **4.** *“In Section 3, it is claimed that, in the early phase of the training, in expectation over the random initialization $\theta \in \Theta$, the TD error is higher when taking the minimum value action than that of a random action. However, this contradicts the experimental results shown in Figure 3, especially Figure 3(a).“*
>
> Please see Figure 6. Figure 6 reports results **across all** the games in the entire Atari benchmark. The results reported in Figure 6 indeed demonstrate that TD error is higher when taking the minimum value action than that of a random action.
>
>
> **5.** *”In Section 4, it is mentioned that the maximum achievable reward in 100 steps is 10. However, the learning curve in Figure 1 (b) is above 10 in the end. How do you get the data used in Figure 1?”*
>
> Thank you for pointing out the typo in the text. The maximum achievable reward in 100 steps is 11. This is due to the fact that the action that transitions the agent to state 3 can be used in state 1 to skip state 2. Thus, the optimal policy in fact can traverse the whole chain in 9 steps.
>
> **6.** *”In Definition 3.2 & 3.5: What is $\hat{a}(s,\theta)$?*
>
> $\hat{a}(s,\theta)$ is any arbitrary action chosen as a function of $s$ and $\theta$. The notation emphasizes the fact that the action choice may depend on both $s$ the current state, and $\theta$ the parameters of the Q-function. That is, we require that the conditions of the definitions hold no matter what action $\hat{a}(s,\theta)$ is chosen as a function of $s$ and $\theta$.

---

> > ### Comment · Reviewer_uoX8 · 2023-11-23
> >
> > Thank you for your reply. Most of my concerns still remain.
> >
> > 1. *The TD error is the loss function used in Q-learning i.e. the goal is to minimize the TD error over all state transitions. By focusing on state transitions where the TD error is higher, larger updates can be made to the Q-function, resulting in faster learning.*
> >
> > I do agree that larger TD errors could lead to larger Q-function updates. However, this does not directly connect with better exploration. More explanations or experimental results are needed to support this.
> >
> > 2. *New experiments for $\epsilon \in [0.01, 0.05]$ in the supplementary material.*
> >
> > I appreciate the new experiments. However, the results are for $\epsilon \in [0.001, 0.005]$, not exactly what I asked for.
> >
> > 3. *In practice $\mathcal{D}(s) − 2\delta − \eta$ is positive as demonstrated by our empirical results on the temporal difference error of MaxMin TD learning versus the -greedy in Figure 3 and Figure 6.*
> >
> > I disagree. What are shown in Figure 3&6 are TD errors, not $\mathcal{D}(s) − 2\delta − \eta$.
> >
> > 4. *The results reported in Figure 6 indeed demonstrate that TD error is higher when taking the minimum value action than that of a random action.*
> >
> > This still does not explain the contradictions between the experimental results in Figure 3 and the theory.

---

> > > ### Author Response · Authors · 2023-11-23
> > >
> > > Thank you for your response.
> > >
> > > **1.** *“I do agree that larger TD errors could lead to larger Q-function updates. However, this does not directly connect with better exploration. More explanations or experimental results are needed to support this.”*
> > >
> > >
> > > It does **indeed** connect with better exploration. In parts of the state-space that have already been frequently visited (i.e. explored), the Q-function will be more accurate i.e. the TD error to be smaller. On the other hand, in parts of the state space that have not been explored, the Q-function will be less accurate i.e. the TD error will be larger. On the experimental side, our results clearly demonstrate faster learning for MaxMin TD learning across the entire Atari benchmark with both double DQN and the distributional Q-learning algorithm QR-DQN.
> > >
> > >
> > >
> > > ---
> > >
> > >
> > > **2.** *”This still does not explain the contradictions between the experimental results in Figure 3 and the theory.”*
> > >
> > > Figure 6 demonstrates that **indeed** the assumption holds **across the entire Atari benchmark**. Figure 3 reports the individual temporal difference error across 12 different games. Amongst these 12 games we only see in 3 games (StarGunner, Bowling and Gravitar) that, although throughout the training the **TD error is indeed higher** just in the starting point we see that TD error is relatively slightly lower. Note that this could also be slightly how these games are structured as also mentioned in the prior work. Nonetheless, even though that is the case in Figure 4 we see that **all of these three games** StarGunner, Bowling and Gravitar indeed **converge to a better policy with MaxMin TD learning**.
> > >
> > >
> > > ---
> > >
> > > **3.** *“I appreciate the new experiments. However, the results are for  $[\epsilon \in 0.001,0.005]$ not exactly what I asked for.”*
> > >
> > > There is a typo here. The results are reported with $[\epsilon \in 0.01,0.05]$. We updated the supplementary material and fixed the typo.
> > >
> > > ---
> > >
> > >
> > > **4.** *“I disagree. What are shown in Figure 3&6 are TD errors, not $\mathcal{D}(s) - 2\delta - \eta$.”*
> > >
> > > If one defines state-dependent versions of $\eta$ and $\delta$ by setting $\eta(s)$ equal to the left-hand side of the inequality in Def 3.1, and $\delta(s,\hat{a})$ equal to the left-hand side of the inequality of Def 3.2, then Proposition 3.4 becomes an equality. That is the expected TD is equal to $\mathbb{E}[\mathcal{D}(s) - \delta(s,a_{min}) - \delta(s,a^*) - \eta(s)]$, where the expectation is over both the random initialization, and over the states encountered in a trajectory. Thus, due to this equality, the results in Figure 3 and 6 (which are expected TD values over states in a trajectory) do **indeed** show that the expected value of this quantity is positive, as in the theoretical results.

---

### Official Review · Reviewer_pNsy · 2023-10-30

**Soundness:** 2 fair
**Presentation:** 2 fair
**Contribution:** 2 fair
**Rating:** 5
**Confidence:** 3

**Summary:**

The work looks at improving sample complexity of deep reinforcement learning (RL) algorithms from the lens of experience collection. A new method based on minimizing state-action value function to increase information gain is proposed. Modifying episilon-greedy, the algorithm leads to more novel experiences by taking actions with the smallest Q-value. Experimentally, the proposed method demonstrates significant improvement in sample complexity in the Arcade Learning Environment, without additional learning parameters.

**Strengths:**

1. The proposed method is well motivated and empirically shows significant improvements in sample efficiency.
2. The paper, in general, is well-structured.

**Weaknesses:**

1. The first few definitions is unclear and unintuitive.
2. The definition of $\hat{a}$ is confusing in Definition 3.2
3. There needs to be a related work section. It is unclear how this approach position among existing works.
4. Figure 1 is too small
5. Missing standard deviation in Figure 3
6. [Minor] the repetition of the questions in conclusion seems like a waste of space to me.

**Questions:**

1. Based on Figure 4, Max-Min TD seems to have higher variance, why is that?
2. Would the method be as effective in sparse-reward setting given that it ties directly to the size of TD?

---

> ### Author Response · Authors · 2023-11-21
> **Author Response**
>
> Thank you for putting in your time to prepare a response for our paper.
>
>
> **1.** *”Would the method be as effective in sparse-reward setting given that it ties directly to the size of TD?”*
>
> The motivating ChainMDP example in Section 4 is indeed **a sparse reward setting**, i.e. only one transition across all $n$ states has non-zero reward, and MaxMin TD learning, as can be seen from the results reported in Section 4, performs substantially better in sparse reward setting as well.
>
>
> **2.** *"The definition of $\hat{a}$ is confusing in Definition 3.2"*
>
> $\hat{a}(s,\theta)$ is any arbitrary action chosen as a function of $s$ and $\theta$. The notation emphasizes the fact that the action choice may depend on both $s$ the current state, and $\theta$ the parameters of the Q-function. That is, we require that the conditions of the definitions hold no matter what action $\hat{a}(s,\theta)$ is chosen as a function of $s$ and $\theta$.

---

### Official Review · Reviewer_zmyD · 2023-10-31

**Soundness:** 2 fair
**Presentation:** 3 good
**Contribution:** 2 fair
**Rating:** 3
**Confidence:** 4

**Summary:**

The paper proposes an exploration method based on minimizing the state-action value function. The method is incorporated into temporal difference based on Q-learning with function approximation. Experiments are conducted using a toy chain MDP and several Arcade Learning Environments. The results are compared to the $\epsilon$-greedy baseline.

**Strengths:**

- The paper addresses the important problem of exploration in reinforcement learning.
- It attempts to provide theoretical justification and analyzes empirical results using toy examples and standard benchmark tasks.

**Weaknesses:**

- Several claims in the paper require further evidence.
- The empirical evaluation lacks detail.
- Details are lacking in addressing the research questions and contributions proposed in the introduction.

**Questions:**

An assumption of the proposed method is that the Q-function, in the initial phase of training, would assign similar values to similar states.
“...early in training the Q-function, on average, will assign approximately similar values to states that are similar…”, “....when the Q-function on average assigns similar maximum values to consecutive states”.
It is unclear how this assumption holds. If a random Q-function processes different consecutive states, then the output value might be arbitrary and not necessarily dependent on the input, even for slightly varied states. Thus, the output could be any random number, not necessarily a similar value.

The text in the plots of Figure 1 is too small and difficult to read. What do each of the plots represent? Are they for different $\epsilon$ values? Which plot corresponds to which value? Also, how does a change in ε affect the results of the proposed MaxMin TD learning?

It is mentioned that "All of the results in the paper are reported with the standard error of the mean”. However, Figure 2 shows the results for the median on the y-axis. Could you clarify what this means?

The claim “.....thus creates novel transitions in exploration with more unique experience collection.” is made, but no evidence is presented in the paper. The results are only compared based on reward performance. How can we be certain that the change in results is due to this particular claim?

In Table 1, the Human Normalized Median is 0.0927 for MaxMin TD and 0.0377 for $\epsilon$-greedy. If 1 is the highest achievable score, then these numbers appear quite low. Do both algorithms fail to learn anything useful? In that case, stating a 248% improvement seems misleading.

What is the QRDQN algorithm baseline in Figure 5? It is not discussed in the paper. What is the difference between $\epsilon$-greedy in Figure 1 and Figure 5? While it is briefly mentioned in the footnote of the supplementary materials, detailed references are not presented.

It is mentioned in the introduction as a contribution that the proposed method "...reaches approximately the same performance level as model-based deep reinforcement learning algorithms," suggesting that the proposed method performs better than model-based. However, no model-based baseline is presented in the experiments, nor is it explained in the text.

---

> ### Author Response · Authors · 2023-11-21
> **Author Response**
>
> Thank you for allocating your time to provide feedback on our paper.
>
> **1.** *”It is unclear how this assumption holds. If a random Q-function processes different consecutive states, then the output value might be arbitrary and not necessarily dependent on the input, even for slightly varied states. Thus, the output could be any random number, not necessarily a similar value.”*
>
> Please see the results reported in Figure 3 and Figure 6. Figure 3 and Figure 6 report the TD error for **both** individual games and **across all** the games in the Atari benchmark, and these results demonstrate that the assumptions **indeed hold** in practice.
>
> **2.** *“The text in the plots of Figure 1 is too small and difficult to read. What do each of the plots represent? Are they for different $\epsilon$ values? Which plot corresponds to which value? Also, how does a change in ε affect the results of the proposed MaxMin TD learning?”*
>
> Yes, the plots in Figure 1 correspond to variations in $\epsilon$. This is already explained in the paper in the last paragraph of Section 4.
>
>
> **3.** *”It is mentioned that "All of the results in the paper are reported with the standard error of the mean”. However, Figure 2 shows the results for the median on the y-axis. Could you clarify what this means?”*
>
> As also mentioned in the caption of the Figure 2, median here refers to human normalized median score. Standard error of the mean means standard deviation.
>
> **4.** *”The claim “.....thus creates novel transitions in exploration with more unique experience collection.” is made, but no evidence is presented in the paper. The results are only compared based on reward performance. How can we be certain that the change in results is due to this particular claim?”*
>
> The evidence for this **is indeed presented in the paper** in Figure 3 and Figure 6. Figure 3 and Figure 6 report the TD error for both individual games and across all the games in the Atari benchmark, and the results reported here demonstrate that MaxMin TD learning results in achieving higher TD error, thus indeed resulting in more novel transitions explored by MaxMin TD learning.
>
> **5.** *”What is the QRDQN algorithm baseline in Figure 5? It is not discussed in the paper.“*
>
> QRDQN algorithm refers to [1].
>
> [1] Distributional Reinforcement Learning with Quantile Regression, AAAI 2018.
>
> **6.** *”It is mentioned in the introduction as a contribution that the proposed method "...reaches approximately the same performance level as model-based deep reinforcement learning algorithms," suggesting that the proposed method performs better than model-based. However, no model-based baseline is presented in the experiments, nor is it explained in the text.”*
>
> Here below we attach the model-based results achieved in the 100K Arcade Learning Environment benchmark. Human normalized median score in the Arcade Learning Environment achieved by SimPLe [1] is 0.144. On the other hand the human normalized median score achieved by MaxMin TD Learning with baseline deep reinforcement learning algorithm QRDQN is **0.158**. Thus, MaxMinTD learning results in achieving higher scores without even building a model of the environment.
>
> [1] Model-Based Reinforcement Learning for Atari, ICLR 2020. **[Spotlight Presentation]**

---

> > ### Comment · Reviewer_zmyD · 2023-11-22
> >
> > Thanks, authors, for the response.
> >
> > Here are the remaining concerns:
> >
> > **1**: Could you please clarify the connection between these empirical results (Figure 3 and Figure 6) and the assumption mentioned here? It is unclear how the empirical results are sufficient to conclude that the assumption holds here. How does the random Q-function work? Does the output of this function depend on the input?
> >
> > **2, 3**: How was the human normalized score calculated? How was the human normalized **median** score calculated? If, on the y-axis, the values are "median" and "80th Percentile" values, then how does the line go in between the standard deviation? It only makes sense if the y-axis in Figure 2 is "mean". It requires further clarification.
> >
> > **4**: As mentioned in the response, could you explain how higher TD error guarantees novel transition in exploration?
> >
> > **6**: If you claim this as a contribution, it needs to be presented in the paper. Merely referring to a paper’s results is not sufficient. Please elaborate on QRDQN. Is it a model-based algorithm? Why can the proposed MaxMin TD Learning (used for Figure 2) not be used directly in comparison with the model-based baseline?
> >
> > Could you respond to this question regarding the results in Table 1?
> >
> > “*In Table 1, the Human Normalized Median is 0.0927 for MaxMin TD and 0.0377 for
> > -greedy. If 1 is the highest achievable score, then these numbers appear quite low. Do both algorithms fail to learn anything useful? In that case, stating a 248% improvement seems misleading.*”

---

> > > ### Author Response · Authors · 2023-11-22
> > >
> > > Thank you for your response.
> > >
> > > **1.** *"Could you please clarify the connection between these empirical results (Figure 3 and Figure 6) and the assumption mentioned here? It is unclear how the empirical results are sufficient to conclude that the assumption holds here. How does the random Q-function work? Does the output of this function depend on the input?"*
> > >
> > > Note that the assumptions here are about the average Q-value, where the average is taken over the different possible random initializations of the Q function. For instance, the randomly initialized Q-function might assign independent, uniformly random values in $[0,1]$ to two consecutive states. Thus any individual random initialization would assign different values to the two states, but on average the value assigned to both states would be $0.5$.
> > >
> > > The theoretical results demonstrate that if the assumption holds, then the temporal difference for MaxMin TD learning will on average be higher than $\epsilon$-greedy. Hence, looking more closely at the proofs of the propositions, it is in fact true that the temporal difference error will on average be equal to the quantity $\mathcal{D}(s) -2\delta -\eta$. Thus, the results reported in Figure 3 and Figure 6 in fact demonstrate that this quantity is positive, as the assumption suggested.
> > >
> > >
> > > **2.** *“How was the human normalized score calculated? How was the human normalized median score calculated? If, on the y-axis, the values are "median" and "80th Percentile" values, then how does the line go in between the standard deviation? It only makes sense if the y-axis in Figure 2 is "mean". It requires further clarification.”*
> > >
> > > Human normalized score is calculated as $HN = (Score_{agent} -Score_{random})/(Score_{human}-Score_{random})$. Note that this is the standard and straightforward normalization that is used in any reinforcement learning paper [1,2,3,4,5]. Figure 2 reports human normalized median across different games, standard deviation across random seeds.
> > >
> > > [1] Human-level Atari 200x faster, ICLR 2023.
> > >
> > > [2] Distributional Reinforcement Learning with Quantile Regression, AAAI 2018.
> > >
> > > [3] Deep Reinforcement Learning with Double Q-Learning, AAAI 2016.
> > >
> > > [4] Dueling Network Architectures for Deep Reinforcement Learning, ICML 2016.
> > >
> > > [5] Human-level control through deep reinforcement learning, Nature 2015.
> > >
> > >
> > >
> > > **3.** *”As mentioned in the response, could you explain how higher TD error guarantees novel transition in exploration?”*
> > >
> > > The TD error is the loss function used in Q-learning i.e. the goal is to minimize the TD error over all state transitions. By focusing on state transitions where the TD error is higher, larger updates can be made to the Q-function, resulting in faster learning. To give an extreme example, if the TD error for a transition is zero, then the gradient of the loss is zero, and using that transition to make an update results in zero change to the Q-function (i.e. zero learning).
> > >
> > > **4.** *“Please elaborate on QRDQN. Is it a model-based algorithm? Why can the proposed MaxMin TD Learning (used for Figure 2) not be used directly in comparison with the model-based baseline?”*
> > >
> > > No, QRDQN is not a model-based algorithm. Please see further detail in [1].
> > >
> > > [1] Distributional Reinforcement Learning with Quantile Regression, AAAI 2018.
> > >
> > > **5.** *“In Table 1, the Human Normalized Median is 0.0927 for MaxMin TD and 0.0377 for $\epsilon$-greedy. If 1 is the highest achievable score, then these numbers appear quite low. Do both algorithms fail to learn anything useful? In that case, stating a 248% improvement seems misleading.”*
> > >
> > > Human normalized score, as explained above, is calculated by the score a human is able to obtain within two hours of play-time with a particular Atari game. Thus, the human normalized median scores here refers to the performance of an algorithm normalized to human score after only 100K interactions, compared to original 200 million frame training.

---

### Official Review · Reviewer_qFHi · 2023-11-07

**Soundness:** 1 poor
**Presentation:** 3 good
**Contribution:** 2 fair
**Rating:** 3
**Confidence:** 4

**Summary:**

This paper proposes a new exploration strategy in reinforcement learning which focuses on taking extremum actions with minimum Q-values. Theoretically, the authors attempt to prove that the TD computed by taking the action with minimum Q-value (denoted as $a_{min}$) is above average (i.e., expected Q-value for a uniform policy) by an amount approximately equal to the disadvantage gap, which is referred to the expected Q-value for a uniform policy minus the Q-value for $a_{min}$. The proposed MaxMin TD Learning policy follows the $\epsilon$-greedy style, where the proposed algorithm takes $a_{min}$ instead of uniform random action for exploration.

**Strengths:**

- This paper proposes an interesting idea of improving the exploration efficiency by taking extremum action, which refers to the action with minimum Q-value.

- The method comes with a nice theoretical motivation, where the authors show the proof of the relationship between TD error inferred by taking $a_{min}$ compared to that for a uniform policy, showing that taking $a_{min}$ as the extremum action more frequently could lead to novel transitions that accelerate learning.

- The proposed method is very general and simple to apply, leading to no additional computational overhead compared to vanilla $\epsilon$-greedy.

- The authors show comparison results with UCB and $\epsilon$-greedy on a toy chain MDP domain and large-scale experimental results by comparing with NoisyNets and $\epsilon$-greedy on Atari 100K.

**Weaknesses:**

- The theoretical contribution of this paper relies on several strong assumptions: (1) expected rewards for a uniform random policy and the $a_{min}$ is $\eta$-uniformed; (2) the Q-value for consequent states $s$ and $s'$ has little difference ($\delta$-smooth); (3) the initialized Q-function results in a policy that is close to uniform random. The main theoretical conclusion that the TD achieved by $a_{min}$ is above-average by an amount approximately equal to the disadvantage gap ($D(s)$) would be wrong if $\delta$ and $\eta$ are not close to 0, because the gap actually equals to $D(s) - 2\delta - \eta$. Also, based on my experience, for value functions parameterized by deep neural nets, the initial policy distribution characterized by the initialized Q-functions is often fairly biased from a uniform distribution. In practice, the value $\epsilon$ would need to gradually decay during the initial phase of training, which means that in practice the theoretically derived conclusion will quickly be invalid in a real training regime.

- In the proposed Algorithm 1, the RL agent always takes $a_{min}$ for exploration action, and no action with intermediate Q-values could be taken for exploration. Unless $a_{min}$ would keep changing among the action set throughout the training, I think the proposed method would easily result in sub-optimal policy compared to $\epsilon$-greedy due to the limited exploration strategy.  For exploration, I'm not convinced it would be generally beneficial to always take a_{min}, and in practice, a_{min} is not guaranteed to always lead to the largest TD error. Though the authors attempt to claim their proposed method is better than UCB through the simple motivating task on chain MDP, I'm still not convinced that the MaxMin TD Learning could generally beat the strong UCB policy variants when tackling challenging RL domains like Atari 2600.

- I think the empirical results on the motivating example are flawed. The authors show the learning curves of MaxMin TD, $\epsilon$-greedy, and UCB, but it is unclear how the exploration policies for the two baselines are specified. For $\epsilon$-greedy, it seems that the authors fix the $\epsilon$ value, otherwise, I expect a well-tuned $\epsilon$-greedy with $\epsilon$ decay properly defined will succeed in the simple chain MDP. It is suspicious why $\epsilon$-greedy converges to a sub-optimal average return. I also wonder if MaxMin TD learning can learn properly without $\epsilon$ decay. It is unfair if the authors allow MaxMin to employ a decayed $\epsilon$, while keeping that for $\epsilon$ or UCB fixed. Please specify the details of each policy.

- For the large-scale Atari 100K evaluation, the baselines are insufficient. As the algorithm focuses on exploration policy, at least it should compare with the UCB-variant of baselines. Also, neither noisy networks nor $\epsilon$-greedy is the SOTA method on Atari 100K. The authors should employ stronger baselines.

- The learning curves for the noisy net are missing in the Atari 100K figures (e.g., Fig 2 and Fig 4). They should be added at least to each game's learning curve.

- It would be more convincing if the authors could evaluate MaxMin TD Learning on a more inclusive range of tasks, e.g., Atari 2600 and mujoco, where the method could work on top of both value-based and policy-based algorithms to verify its generality.

**Questions:**

Please refer to the WEAKNESSES section.

---

> ### Author Response · Authors · 2023-11-21
> **Author Response Part I**
>
> Thank you for spending your time to write a response to our paper.
>
> **1.** *“For the large-scale Atari 100K evaluation, the baselines are insufficient. As the algorithm focuses on exploration policy, at least it should compare with the UCB-variant of baselines. Also, neither noisy networks nor $\epsilon$-greedy is the SOTA method on Atari 100K. The authors should employ stronger baselines.”*
>
> MaxMin TD Learning is a zero-cost algorithm to boost TD learning. Note that $\epsilon$-greedy is currently the main method used in deep reinforcement learning due to its efficiency and effectiveness [1,2,3,4,5]. Even **the best performing algorithm in Atari 100K** that was quite recently published in ICML 2023 [5] is indeed using $\epsilon$-greedy. Nonetheless, although MaxMin TD learning is a zero-cost technique we still further provided comparisons against more expensive baseline techniques such as NoisyNetworks.
>
> [1] Human-level Atari 200x faster, ICLR 2023.
>
> [2] Bootstrapped Meta-Learning, ICLR 2022. [Oral Presentation]
>
> [3] Combining Q-learning and search with amortized value estimates. ICLR, 2020.
>
> [4] Reincarnating Reinforcement Learning:Reusing Prior Computation to Accelerate Progress, NeurIPS 2022.
>
> [5] Bigger, Better, Faster: Human-level Atari with human-level efficiency, **ICML 2023**.
>
>
>
> **2.** *”It would be more convincing if the authors could evaluate MaxMin TD Learning on a more inclusive range of tasks, e.g., Atari 2600 and mujoco, where the method could work on top of both value-based and policy-based algorithms to verify its generality.”*
>
> The focus of our paper is **high-dimensional state representation MDPs** which follows the line of work [1,2,3,4,5,6,7,8] (i.e. Arcade Learning Environment). Furthermore, currently our paper already reports results both in Atari 100K and Atari 2600. These results are reported in Figure 2, Figure 5 and Figure 4 respectively.
>
> [1] Human-level Atari 200x faster, ICLR 2023.
>
> [2] Distributional Reinforcement Learning with Quantile Regression, AAAI 2018.
>
> [3] Deep Reinforcement Learning with Double Q-Learning, AAAI 2016.
>
> [4] Dueling Network Architectures for Deep Reinforcement Learning, ICML 2016.
>
> [5] Deep Reinforcement Learning at the Edge of the Statistical Precipice, NeurIPS 2021. [Oral Presentation]
>
> [6] Model based reinforcement learning for Atari. ICLR 2020. [Spotlight Presentation]
>
> [7] Emphatic Algorithms for Deep Reinforcement Learning, ICML 2021.
>
> [8] Implicit Quantile Networks for Distributional Reinforcement Learning, ICML 2018.
>
>
> **3.** *“I think the proposed method would easily result in sub-optimal policy compared to $\espilon$-greedy due to the limited exploration strategy.“*
>
> Please see the results reported across the entire Atari 100K benchmark in Figure 2 and Figure 5. These results indeed demonstrate that MaxMin TD learning **does not** result in a sub-optimal policy compared to $\epsilon$-greedy.

---

> > ### Author Response · Authors · 2023-11-21
> > **Author Response Part II**
> >
> > **4.** *“For exploration, I'm not convinced it would be generally beneficial to always take a_{min}, and in practice, a_{min} is not guaranteed to always lead to the largest TD error.”*
> >
> > Please see the results reported in Figure 3 and Figure 6. Figure 3 and Figure 6 already report the TD error for both individual games and across all the games in the entire Atari 100K benchmark, and these results demonstrate that **indeed MaxMin TD learning results in larger TD error**.
> >
> >
> > **5.** *“I think the empirical results on the motivating example are flawed. The authors show the learning curves of MaxMin TD, $\epsilon$-greedy, and UCB, but it is unclear how the exploration policies for the two baselines are specified. For $\epsilon$-greedy, it seems that the authors fix the $\epsilon$-value, otherwise, I expect a well-tuned $\epsilon$-greedy with $\epsilon$ decay properly defined will succeed in the simple chain MDP. It is suspicious why $\epsilon$-greedy converges to a sub-optimal average return. I also wonder if MaxMin TD learning can learn properly without $\epsilon$ decay. It is unfair if the authors allow MaxMin to employ a decayed $\epsilon$, while keeping that for $\epsilon$ or UCB fixed. Please specify the details of each policy.”*
> >
> > It is **incorrect** to expect for $\epsilon$-greedy to perform well in the chain MDP. To see why, observe that for a policy that acts randomly in each step, there is a probability of $3/4$ of resetting to one of the first 3 states in the chain. Thus, the probability of such a policy reaching the end of the chain is less than $1/4^7 < 1/16,000$. Unlike UCB or MaxMin TD learning, $\epsilon$-greedy exploration does not in any way take into account which states have been visited before when deciding to take an action, and thus will require many more steps to reach the end of the chain for the first time and receive a non-zero reward signal. Also note that the value of $\epsilon$ is fixed for both $\epsilon$-greedy and MaxMin TD learning without any decay. Furthermore, note also that both of these models are trained for 10K steps. Thus, with many more steps $\epsilon$-greedy will eventually converge, albeit much more slowly than MaxMin Novelty.
> >
> >
> > **6.** “In practice, the value $\epsilon$ would need to gradually decay during the initial phase of training, which means that in practice the theoretically derived conclusion will quickly be invalid in a real training regime.“
> >
> > Again please see the results reported in Figure 3 and Figure 6. Figure 3 and Figure 6 already report the TD error for both individual games and across all the games in the entire Atari 100K benchmark, and these results demonstrate that the theoretical derived results are **indeed valid in practice**.

---

### Meta-Review · Area_Chair_P58J · 2023-12-04

**Metareview:**

This paper introduces a new method for RL that focuses on experience collection in high-dimensional complex MDPs considering experiences obtained through extremum actions.

**Reviewers have reported the following strengths:**

- Novel and interesting idea;
- Simplicity of the method;
- Quality of writing.

**Reviewers have reported the following weaknesses:**

- Empirical evaluation;
- Number and nature of the necessary assumptions.

**Decision**

This paper received unanimous negative reviews. The Reviewers agree that the paper needs an improved empirical evaluation. Moreover, the theoretical analysis seems to rely on excessively strong assumptions, which can limit the method's applicability.

**Justification For Why Not Higher Score:**

N/A

**Justification For Why Not Lower Score:**

N/A

---

### Decision · Program_Chairs · 2024-01-16

Reject